# Looking for the Ideal Probiotic Healing Regime

**DOI:** 10.3390/nu15133055

**Published:** 2023-07-06

**Authors:** Alexandra Menni, Moysis Moysidis, Georgios Tzikos, George Stavrou, Joulia K. Tsetis, Anne D. Shrewsbury, Eirini Filidou, Katerina Kotzampassi

**Affiliations:** 1Department of Surgery, Aristotle University of Thessaloniki, 54636 Thessaloniki, Greecemoisisdoc@gmail.com (M.M.); tzikos_giorgos@outlook.com (G.T.); a_shrewsbury@yahoo.com (A.D.S.); 2Department of Colorectal Surgery, Addenbrooke’s Hospital, Cambridge CB2 2QQ, UK; stavgd@gmail.com; 3Uni-Pharma S.A., 14564 Athens, Greece; jtsetis@uni-pharma.gr; 4Laboratory of Pharmacology, Faculty of Medicine, Democritus University of Thrace, 68100 Alexandroupolis, Greece; efilidou@hotmail.com

**Keywords:** wound healing, *Lactoplantibacillus plantarum*, *Lacticaseibacillus rhamnosus*, *Lactobacillus acidophilus*, *Bifidobacterium longum*, *Lacticaseibacillus casei*, *Limosilactobacillus reuteri*, *Lactobacillus fermentum*, *Levilactobacillus brevis*, *Saccharomyces boulardi*

## Abstract

Wound healing is a multi-factorial response to tissue injury, aiming to restore tissue continuity. Numerous recent experimental and clinical studies clearly indicate that probiotics are applied topically to promote the wound-healing process. However, the precise mechanism by which they contribute to healing is not yet clear. Each strain appears to exert a distinctive, even multi-factorial action on different phases of the healing process. Given that a multi-probiotic formula exerts better results than a single strain, the pharmaceutical industry has embarked on a race for the production of a formulation containing a combination of probiotics capable of playing a role in all the phases of the healing process. Hence, the object of this review is to describe what is known to date of the distinctive mechanisms of each of the most studied probiotic strains in order to further facilitate research toward the development of combinations of strains and doses, covering the whole spectrum of healing. Eleven probiotic species have been analyzed, the only criterion of inclusion being a minimum of two published research articles.

## 1. Introduction

Trauma, of whatever cause, either a clean surgical wound or a severe traumatic injury, leads to an alteration in the beta diversity of resident microbiota due to the oxidative stress caused by both tissue injury and anesthesia. The alteration of the microbiota involves both the intestinal microbiota—being the most studied and definitely considered as a generalized reaction—and the local microbiota, in this case, that of the skin [1,2,3,4].

Wound healing is an evolutionary, complex, curative response to tissue injury, aiming to restore tissue continuity and resistance to externally applied forces, in an overlapping consequence of three phases: Initially, the blood clot plugs the bottom of the wound and acts as a temporary barrier to external invasion while, in parallel, coagulation factors indirectly involved in neutrophil and macrophage migration to initiate the inflammatory phase—practically to clear the field. When this is complete, macrophages turn into M2 anti-inflammatory macrophages to guide tissue regeneration in parallel with fibroblast migration. Consequently, a granulation tissue consisting of macrophages, fibroblasts, connective tissue, and new blood vessels is developed to replace the initial blood clot, after which fibroblasts differentiate into α-SMA-expressing myofibroblasts to increase the wound stiffness, thus terminating tissue remodeling. Finally, wound cellularity decreases, and collagen type III in the extracellular matrix is replaced by collagen type I to form the scar [5,6]. Although it is largely a “standard” process, in recent years, there has been increasing certainty that the disruption of the local microbiome, as a consequence of trauma, is involved in the prolongation of the healing time the most simple, the biofilm formation and the growth of pathogenic microorganisms, to the most complex, the prolongation of the inflammatory phase and the transition from acute to delayed healing; and, even in clinical cases generally, to bacterial colonization, and possibly clinical sepsis [7,8].

According to the current definition endorsed by both the Food and Agriculture Organization of the United Nations (FAO) and the World Health Organization (WHO), as well as the International Scientific Association for Probiotics and Prebiotics (ISAPP), “probiotics are live microorganisms that, when administered in adequate amounts, confer a health effect on the host” [9]. Postbiotics, on the other hand, are the metabolic products that are secreted by probiotics, such as cell-free supernatants [10], and may also have a favorable effect on the host, while parabiotics are the inactivated remnants of probiotics, such as cell-lysates and heat-killed probiotics, which also have beneficial effects [11]. In vitro and in vivo studies have already indicated that probiotics and postbiotics exert potential suppressive effects on various infectious, immune-related, and inflammatory conditions. Although prevention of wound infection may not have a direct effect on wound healing, it seems to play a significant indirect role, as, ultimately, an infected wound requires more time to heal than a non-infected one. They have the ability to control inflammation and induce local and systemic immune responses, while various positive stimulatory effects on wound healing have been reported [12,13,14,15]. For these reasons, there is a progressive trend towards local probiotic therapy to promote healing by restoring the affected microbiome. There are several studies on cell line cultures as well as in experimental wound models, mainly in rats, reporting on cytokines and other molecules involved in the complex mechanism of overlapping phases of the healing process. All, or almost all, of these studies conclude that the probiotic, or the combination formula of probiotics under investigation, promotes “wound healing” by affecting all three stages of the wound healing process (Figure 1). However, it is very difficult to assess the precise involvement of probiotics or postbiotics in the healing process since each strain has separate, distinctive, and even multi-factorial features of action [16,17].

With this knowledge, and given that a multi-probiotic formula exerts better results than a single strain [16,18,19,20], although there are references to the opposite [21,22,23], the pharmaceutical industry has embarked on a race to produce a complex formulation containing a combination of probiotics, capable not only of modulating all the healing phases but also without the one probiotic inhibiting the action of the other [19,20,24,25,26]. Hence, the object of this review is to describe what is known to date of the distinctive and even multi-factorial “healing” mechanisms of each probiotic strain in order to further facilitate research for the development of combinations of strains and doses covering the whole spectrum of healing. Eleven of the most studied probiotic species were analyzed in the current review, the only criterion of inclusion being a minimum of two published research articles. As most of these studies were performed in vitro and in vivo, the results discussed and analyzed hereafter mainly focus on the pre-clinical level.

## 2. “Healing” Mechanisms of Probiotics

### 2.1. Lactiplantibacillus plantarum 

*Lactiplantibacillus plantarum* (Formerly *Lactobacillus plantarum*) [27] is one of the most studied probiotic bacteria [28]. A large part of the research has been conducted in an almost standardized excisional wound model, mainly in rats, while the contribution of keratinocytes or other cell line cultures is also of great value.

Lactic acid bacteria are the main and most important subgroup of probiotics found in dairy products; this is why early research on probiotics began from the use of traditional dairy fermentation products, such as cheese or yogurt, or, later, probiotic strains isolated from these cultures. Nasrabadi et al. from Iran investigated the effect of an *L. plantarum* strain isolated from traditional Iranian cheese on 60% acetic acid-induced gastric ulcers in rats. *L. plantarum,* given intragastrically at a dose of 10^10^ cfu/day, was found to reduce the ulcer area and enhance gastric healing by day 4, as compared to controls, the most significant reduction being observed on day 7. In parallel, neutrophils were greatly increased on day 3 and then decreased by day 5, while macrophages and fibroblasts significantly increased on days 3 to 5 and decreased on day 7—all findings compared to the control treatment. This is probably the first recognition and report that *L. plantarum* could significantly affect the inflammatory phase of healing via stimulating the immune system, with an influx of neutrophils and macrophages in the early phase and then elimination in the late phase of the healing process, while the influx of fibroblasts on day 7 suggests the collagen deposition phase initiating earlier than in controls [29].

The previous findings were again documented by the same research group in two quite similar studies performed on an excisional cutaneous wound model in rats after applying *L. plantarum* topically for 21 days. A significant reduction in local inflammation, an acceleration of wound healing, and an earlier increase of fibroblasts in *L. plantarum*-treated rats were prominent compared to control-treated rats [30,31].

Besides the anti-inflammatory properties of *L. plantarum* found in cutaneous excisional trauma, the *L. plantarum* 299v topical effects on infected with Pseudomonas aeruginosa burn-trauma in rats was also investigated in a 14 d scheduled treatment. First of all, *L. plantarum* 299v exhibited, in vitro, the highest inhibitory diameter (16 mm) against resistant *P. aeruginosa* compared to other probiotics tested. Although this observation may not be a direct effect on wound healing, the preventive effect of *P. aeruginosa*’s infection seems to also ultimately play a role in wound healing. Specifically, the percentage of wound healing on day 14 was significantly higher in the *L. plantarum* group compared to the control or imipenem-treated; the mean size of the wound on day 14 was significantly reduced in the probiotic group versus the control group; the wound cultures revealed a significantly increased percentage with no pathogen growth; and the mean leukocyte count was significantly higher in the probiotic group. When viable probiotic cells and supernatant-only treatments were compared, the viable probiotic cells treatment was inferior [32].

In the same manner, in a rabbit model of full-thickness burn wounds, the topical therapy with *L. plantarum* resulted in a significant (almost 50%) inhibition of type I collagen mRNA expression, as well as a similar reduction in total collagen protein production, expressed as tissue hydroxyproline content; these findings were also verified by means of direct histochemical visualization and quantification of collagen deposits. Moreover, it is of interest to mention that *L. plantarum*-treated burn wounds showed a markedly greater relative abundance of thinner green fibrils, consistent with greater amounts of immature type III collagen; these indicate that probiotic therapy can modulate not only the quantity but also the type of collagen synthesized in response to injury and infection, thus alleviating excessive scarring [13].

The cell-free supernatant and protein-rich fraction postbiotic from *L. plantarum* USM8613 was found to reduce the number of viable *Staphylococcus aureus* cells, as well as biofilm thickness both in vitro and in vivo in *S. aureus*-infected porcine skins. These findings were attributed, according to genome analysis, to the production of several plantaricins, especially the plantaricins EF and JK, which enhanced the anti-staphylococcal effects of *L. plantarum* USM8613. Again, these anti-microbial effects of the postbiotic from *L. plantarum* USM8613 may also indirectly favor wound healing. Regarding the healing process, in a rat excisional wound model, *L. plantarum* USM8613 postbiotic protein fraction significantly enhanced the production of cytokines and chemokines at wound sites through the consecutive stages of wound recovery: β-defensin, IL-4, IL-6, and TGF-β were found significantly up-regulated up to day 4 and then decreased to lower levels in relation to controls. IFN-γ was increased after day 1 and decreased over time in both groups, but in the probiotic-treated group, the decrease was greater by day 8—not until day 12 in controls—and matrix metalloproteinases (MMPs) gradually reduced over time in the probiotic-treated animals, but not in controls. All these findings were compatible with the macroscopic and histological findings of healing: enhanced wound contraction percentage of 54% by day 4, while controls achieved a wound contraction percentage of above 50% by day 8; accelerated keratinocyte migration towards the center of the wound and complete re-epithelization by day 12, in relation of controls which achieve complete re-epithelization only by day 16; the presence of inflammatory cells at wound sites mostly disappeared on day 12 as compared to controls on day 16 [33].

*L. plantarum*-treated diabetic rats subjected to a square (1.5 × 1.5 cm) dorsal wound presented with an increased rate of wound closure from day 3 in relation to control-treated diabetic rats; and finally, on day 14, more than 95% of the wound area was closed in relation to the control-treated. These findings are consistent with that of gene expression of pro- and anti-inflammatory cytokines: topical treatment with *L. plantarum* markedly decreased the expression of IL-1β and TNFα on day 7, which further down-regulated up to day 14; on the other hand, a significantly increased expression of the anti-inflammatory mediators IL-10 and TGFβ1 was prominent up to day 7, with greater expression up to day 14 in relation to control-treated rats [34].

At the clinical level, in a recent RCT on 22 chronic diabetic foot ulcers patients, the topical application of *L. plantarum* demonstrated a significant reduction in ulcer area, and a significantly increased fibroplasia and angiogenesis at 21 days, while Masson’s and α-smooth muscle actin staining demonstrated a significantly improved cellular proliferation, as well as increases in collagen, myofibroblasts, and micro-vessel density, in relation to control treatment. Additionally, it is of exceptional interest to mention a switch in macrophage phenotype from pro-inflammatory M1 (CD68) to M2 (CD163) anti-inflammatory and reparative cells, beginning from day 7 to day 14, the process significantly speeding up in *L. plantarum*-treated patients [35].

The healing properties of *Lactobacillus plantarum*, when given orally at a dose of 10^6^ cfu/g, were evaluated in excisional, re-sutured incision and dead space wounds in rats in comparison to control and to *L. acidophilus* treatment. The 2.5 cm excision wound performed in the neck exhibited a significantly lower mean percentage closure and a longer mean time for complete epithelization when treated with *L. plantarum,* as compared to *L*. *acidophilus*, the findings being closer to control values. Re-sutured incisions and dead space wounds also showed a significantly lower wound breaking strength, and granuloma dry weight, as compared to *L. acidophilus*, and were very similar results to control. To the best of our knowledge, this is the only study in which the *L. plantarum* healing properties were found similar to control treatment; this is probably attributable to the low concentration of formula given orally (10^6^ cfu/g), the fact that treatment lasted 10 days, and the lack of specification of the strain used [36]. Similar doses are reported in different studies referred to in this review [18,29].

An explanation for the above issues may also be found in the paper by Coelho-Roche et al. [37]. They aimed to investigate the probiotic properties and the gastrointestinal protective effects of nine novel *L. plantarum* strains isolated in Brazil. All *L. plantarum* strains exhibited good tolerance to bile salts and low pH and were able to inhibit common gastrointestinal pathogens. Seven out of the nine strains tested exhibited a protective effect against histopathological and inflammatory damage induced by 5-fluorouracil, while all nine strains reduced eosinophilic and neutrophilic infiltration. Gene expression analysis of inflammatory markers showed that five strains up-regulated IL-10, while four others down-regulated both IL-6 and IL-1b. Two strains also exhibited high adhesion rates to Caco-2 cells, while most strains presented a phenotypical resistance to aminoglycosides, vancomycin, and tetracycline. No strain showed hemolytic or mucolytic properties.

The topical effect of *L. plantarum* UBLP-40 on a standardized excisional wound in rats regarding the mRNA expression of pro-inflammatory, healing, and angiogenetic factors involved in the healing process was recently investigated. *L. plantarum* UBLP-40 exhibited a strong anti-inflammatory effect, as it was found to reduce the mRNA expression of IL-1β, with the more profound, statistically significant results obtained by the end of the experiment, on day 15 and of IL-6 mRNA expression by day 8 compared to controls. It was apparent that it significantly promoted the initiation of the healing process, while later, it contributed to the attenuation of healing factors, possibly reducing hypertrophic wound scar formation. Thus, it resulted in a mild but statistically significant induction of collagen type III mRNA expression from day 2 while reducing the TGF-β expression on day 8 compared to controls. In parallel, it up-regulated the EGF mRNA throughout the study period and down-regulated the VEGF mRNA up to day 8 in relation to controls [19].

The topical effect of 10^11^ cfu/gr of *L. plantarum* UBLP-40 was compared with that of a combined formula of *L. rhamnosus* UBLR-58 and *B. longum* UBBL-64 and with placebo treatment, on six excisional wounds of 8 mm, on the dorsal skin of rats. Wound area healing, expressed as the remaining percentage of trauma, was assessed by means of photography and image analysis software throughout the healing period: on day 4, *L. plantarum* led to an almost half reduction of wounded area (58.8%) in relation to combo-formula (70.5%) and the placebo (87.8%), thus confirming the strong anti-inflammatory properties of *L. plantarum*; on day 8, *L. plantarum* seemed to have reduced the speed of the healing process in the proliferation phase (from 58.8% to 41%) in relation to the combo-formula which sped up healing (from 70.5% to 22.8%), while the placebo reached 49%; on day 16, *L. plantarum* seemed to reduce the wounded area to 6.2%, in relation to the combo-formula which expressed the perfect healing effect (3.9%), while the placebo reached only 25% [18].

The parabiotic lysate of *L. plantarum* SGL07, applied on scratched keratinocytes monolayer culture, was found to significantly accelerate re-epithelialization in keratinocyte monolayers and exhibit a strong induction of keratinocyte migration and proliferation versus control treatment. Moreover, after a 24 h exposure of keratinocytes to parabiotic *L. plantarum* SGL07 lysate, a significant decrease of MCP-1, RANTES, and IL-8 release was prominent, thus documenting its anti-inflammatory effect [38].

The rapid repair of keratinocyte scratch assay after treatment with the postbiotic soluble fraction of *L. plantarum* Lp-115 lysate was attributed to the ability of postbiotics to increase the nitrate levels in culture medium in relation to control since a marked and significant up-regulation of nitric oxide synthase 2 (NOS2) expression was demonstrated by immunoblotting data and by nitrite level assay [39].

Tsai et al. reported that parabiotic heat-killed *L. plantarum* GMNL-6, when applied on human foreskin fibroblasts (Hs68 cells), promotes collagen synthesis and up-regulates the gene expression of serine palmitoyl-transferase small subunit A, compared to control treatment. Furthermore, it exerts an inhibitory effect on the expression of αSMA, Smad2/3, and phosphorylated Smad2, as indexes of TGF-β induced myofibroblast differentiation, in a dose-dependent manner. Additionally, it accelerates a positive effect—up-regulation of MMP1 expression in the early phase of healing on day 5, but gradually decreases in the next phase on day 9, in parallel with the presence of structured collagen fibers in the dermis of treated wounds, as well as decreased expression of αSMA. All the above strongly indicate that parabiotic (heat-killed) *L. plantarum* GMNL-6 not only promotes apparent wound healing but also prevents excessive skin fibrosis, all mediated by the functional ingredient of the probiotic cell wall, the lipoteichoic acid [40].

In the same manner, Dubey et al. [41] assessed the wound-healing properties of *L. plantarum* MTCC 2621. The postbiotic cell-free supernatant from *L. plantarum* MTCC 2621 applied on a scratched epithelial A549 cell line in different concentrations (6.25% and 12.5%) was found to significantly increase the percentage of wound healing during the 24 h treatment compared to controls. Then, in an excisional wound model in mice, topical application of *L. plantarum* in a gel form exhibited considerably increased wound healing to that of the control, as assessed by the percent reduction of the wound area. Histopathology on day 7 revealed an enhanced proliferation of fibroblasts, angiogenesis, re-epithelization, collagen deposition, and granulation tissue as compared to controls, while wounds were totally healed on day 14 versus incomplete healing in controls. Similar wound-healing activity was observed in wounds infected with *Staphylococcus aureus*. *L. plantarum*-treated and infected excisional wounds depict faster re-epithelization with a reduced infiltration of leukocytes, increased fibroblastic activity, and increased collagen deposition compared to the control treatment. Finally, *L. plantarum* treatment resulted in the up-regulation of serum levels of pro-inflammatory cytokine IL-6 in the early phase of wound healing and of IL-10 in the later phase in relation to controls [41].

Finally, another option for research is performed in human intestinal subepithelial myofibroblast cultures, thus resembling intestinal mucosa healing, as occurs in inflammatory bowel diseases. The stimulation of myofibroblasts with *L. plantarum* UBLP 40 resulted in the up-regulation of the mRNA expression of only the chemokines CXCL10 and CXCL8, with no significant effect on CXCL1, CXCL2, CXCL4, CXCL2, and CCL5. However, it was found to regulate wound healing-related factors since it increased the mRNA expression of collagen type III and of fibronectin; at a higher dose (10^4^ cfu/mL]) of collagen type I; at a low dose of 10^2^ cfu/mL, it led to a statistically significant increase of Tissue Factor mRNA. Finally, negative results were obtained regarding α-SMA mRNA. Moreover, *L. plantarum* UBLP 40 promotes the wound-healing process through the induction of myofibroblast migration: both 10^2^ cfu/mL and 10^4^ cfu/mL doses were found to significantly increase the migratory rate at 24 h, while the higher dose maintained higher migration after 48 h [20].

### 2.2. Lacticaseibacillus rhamnosus (Formerly Lactobacillus rhamnosus) 

As early as 2006, the probiotic *L. rhamnosus GG* ATCC 53103 was found to synthesize an acid and heat-stable, low-molecular-weight postbiotic peptide, thus posing the ability to inducing cytoprotective heat shock proteins (Hsp) in murine intestinal epithelial cells, in a time- and concentration-dependent manner, involving transcriptional regulation by the transcription factor HSF-1. Of further interest, *L. rhamnosus GG* not only provides protection against oxidant stress by up-regulating the Hsp25 and Hsp72 starting at 18 and 4 h, respectively, but also modulates signal transduction pathways by activating mitogen-activated protein kinases (MAPKs). Furthermore, it promotes the protection of intestinal epithelial cells from oxidant stress by improving their viability in the oxidative stress phase through the maintenance of F-actin, protecting the cytoskeleton from oxidative radicals [42].

In a Swiss mice model of excisional wounds, *L. rhamnosus* CGMCC 1.3724 LPR was given orally from recovery from anaesthesia until 14 days thereafter. *L. rhamnosus* exhibited, macroscopically, a faster wound closure rate compared to the control group since it was histologically found to stimulate the re-epithelization process (an increase of epithelial tongue length and of PCNA-positive keratinocytes and reduction of epithelial gaps), enhance angiogenesis (increased blood vessels density—VEGF levels and blood flow laser Doppler measurements), reduce wound inflammation (reduced granulation tissue and leukocytes infiltration, reduce NAG activity and CCL-2 chemokine levels, and mast cells density), and reduce collagen deposition and scar area [43].

As seen earlier with *L. reuteri* [44], the same research group tested *L. rhamnosus* GG ATCC 53103 for whether it inhibits the toxic effects of *S. aureus* on epidermal keratinocytes culture. A co-culture of keratinocytes with the pathogen *S. aureus* and *L. rhamnosus* GG exhibited a significantly higher viability than in monolayers infected with the pathogen alone; this occurred by either the probiotic lysate or spent culture fluid, the maximum protection achieved when keratinocyte monolayers were exposed to *L. rhamnosus* GG 2 h prior to infection. However, live probiotic bacteria alone do protect, even up to 12 h later. Additionally, live *L. rhamnosus* bacteria protect from *S. aureus* pathogen, not only by means of inhibition of *S. aureus* adhesion by competitive exclusion as occurs with *L. reuteri* [44] but also by direct inhibition of *S. aureus* growth, thus increasing the viability of the infected keratinocytes by 57% [45]. Once again, it should be noted that the anti-microbial properties of *L. rhamnosus* may not have a direct effect on wound healing but rather an indirect one by inhibiting further infections.

The topical effect of *L. rhamnosus UBLR-58* regarding the mRNA expression of pro-inflammatory, healing, and angiogenetic factors involved in the healing process in relation to *L. plantarum* UBLP-40 was recently investigated on a standardized excisional wound in rats [19]. A statistically significant increase in the expression of IL-1β on days 2 and 15 and of IL-6 throughout the 14 d healing period compared to *L. plantarum* was found, which supports the hypothesis that *L. rhamnosus* exerts a less active role in the progression of the inflammatory phase compared to *L. plantarum*. However, *L. rhamnosus* presented as more active in the wound healing and angiogenesis process, significantly increasing the mRNA expression of TGF-β and of α-SMA. However, there was a less profound up-regulation of collagen type III mRNA expression in relation to *L. plantarum* and a significant up-regulation of both EGF and VEGF mRNA expression from day 2, with no further increase over time in relation to *L. plantarum.*

When the parabiotic *L. rhamnosus* GG ATCC 53103 lysate was added in a scratched keratinocyte monolayer, it was found to significantly stimulate cell migration by 7-fold and cell proliferation by 2-fold in relation to controls. Moreover, in the presence of Mitocycin C, a well-known cell proliferation inhibitor, re-epithelialization was little affected (10% less) in relation to a similar co-culture of *L. reuteri* plus Mitocycin C. This finding suggests that the parabiotic *L. rhamnosus* GG was more effective in stimulating keratinocyte migration than proliferation, the underlining mechanism being the increase in expression of CXCL2 chemokine and its receptor, CXCR2 [45].

Finally, the probiotic strain *L. rhamnosus* LR-32 was tested to determine whether it might control inflammation by altering the inflammatory phenotype of the human CD14+ monocytes culture. *L. rhamnosus* was found to only slightly increase the TNF-α and CXCL-8 cytokines levels, but not IL-1Β and IL-6 levels, in the culture supernatant compared to that of non-challenged cells. When compared, the effects of co-culture with *L. rhamnosus* or with *L. acidophilus,* with respect to the expression of genes involved in apoptosis, as well as to the mitogen-activated protein kinases (MAPKs), and to the family of protein kinases and the genes with any anti-microbial characteristics, *L. rhamnosus* generally exhibited a very low or even negative expression value in relation to the *L. acidophilus*-treated [46].

### 2.3. Lactobacillus acidophilus

*Lactobacillus acidophilus’s* healing properties, when given orally, were first studied comparatively with those of *L. plantarum* in excisional, re-sutured incision, and dead space wounds. Excision wounds in rats’ necks, having a mean diameter of 2.5 cm, exhibited a significantly higher mean percentage closure as well as a shorter mean time to complete epithelization when treated with *L. acidophilus*, as compared to *L. plantarum* and control, at days 4, 8, 12, 16, and 18. The *L. acidophilus* group re-sutured incision and dead space wounds also showed a significantly higher breaking strength and granuloma dry weight compared to *L. plantarum* and controls; microscopically, an increase in granulation tissue, with increased cell infiltration with lymphocytes and macrophages, and a moderate increase in collagen and hydroxyproline content were prominent in the same group. Of interest, the *L. acidophilus* group showed stellate-shaped scars compared to oval scars in the controls, the finding attributed by the authors to increased wound contraction due to enhanced granulation tissue formation as observed in a dead space wound model [36].

The re-epithelialization potential of the postbiotic soluble fraction of *L. acidophilus* lysate was tested on a wound (scratch test) human keratinocyte HaCaT cell-monolayer model. The postbiotic of *L. acidophilus* was found to significantly accelerate the rate of the monolayer repair process, in relation to untreated controls, at 20 and 28 h post-injury. This seems to occur through a marked and significant up-regulation of NOS2 protein expression, inducing a significant increase of nitrite levels in the culture medium. The positive, highly statistically significant correlation between NOS2 protein levels, as well as nitrite levels, and the % wound re-epithelialization findings strongly suggest that—given the involvement of NO in the wound repair process—the soluble fraction of postbiotic up-modulates the NOS2 expression and thus plays a crucial role in the re-epithelialization process. Of note, comparison among *L. plantarum*, *S. thermophilus*, and *L. acidophilus* treatments to induce an increase of nitrite levels and thus to up-modulate NOS2 expression revealed a significant difference in relation to controls, the *L. acidophilus* being the worst (*p* < 0.01) and *L. plantarum* being the best (*p* < 0.0001) [39].

Next to them, Vale et al. [46] investigated the gene transcription of the antibacterial response of monocytes after exposure to *L. acidophilus* LA-5. CD14+ monocytes obtained from the peripheral blood mononuclear cells of healthy donors after being subjected to 12 h co-culture with *L. acidophilus* were found to up-regulate the genes of IL-12B, IL-1Β, TNF-α, CXCL-8, and TLR-2 and increase the TNF-α, CXCL-8, IL-1Β and IL-6 cytokines production levels in the supernatant compared to the non-challenged cells. *L. acidophilus* treatment also showed higher values in all MAPKs analyzed, as also occurred with the genes involved with apoptosis, such as the XIAP, PYCARD, and RIPK1, and the genes of the protein kinases family, such as CHUK, IKBKB, and RAC1. Finally, genes having anti-microbial characteristics, such as LYZ and TICAM, followed the same transcriptional trend as controls, with all findings supporting the belief that *L. acidophilus* LA-5 significantly promoted the antibacterial activity of innate defense in tissues and, thus, indirectly contribute to the enhancement of the wound-healing process.

Recently, Tarapatzi et al. [20] investigated the same probiotic strain, *L. acidophilus* LA-5, in relation to the expression of chemokines and wound-healing related factors, as well as the migratory rate of human colonic subepithelial myofibroblasts in culture. The stimulation of myofibroblasts with *L. acidophilus* was found to decrease the mRNA expression of CΧCL2, CXCL6, and CXCL8 chemokines but to significantly up-regulate the CXCL10 mRNA and to exert no impact on the CXCL1, CXCL4, CCL2, and CCL5 mRNAs. Moreover, it decreased the mRNA expression of collagen types I and ΙΙΙ but had no effect on its total protein production. Fibronectin and Tissue Factor were found down-regulated in a dose-depended manner, while no effect on α-SMA mRNA expression was prominent. Finally, *L. acidophilus*, independently of the dose, expressed no significant migratory effect [20].

### 2.4. Levilactobacillus brevis (Formerly Lactobacillus brevis) 

The probiotic *Lactobacillus brevis* GQ423768, isolated from traditional Iranian dairy products, was tested for its healing effects on a cutaneous, full-thickness, open excision wound of approximately 1.5 × 1.5 cm on rats. On day 7, the wounds treated with *L. brevis* were found to be significantly smaller than in the controls. Histologically, a significant reduction in the number of neutrophils and an increase in macrophages and fibroblasts from day 3 and thereafter was clear; the number of the latter being reduced after the 14th day of treatment in relation to controls suggests that *L. brevis* exert anti-inflammatory properties [47].

The previous findings were also documented by the same research group in a similar study performed on an excisional cutaneous wound model in rats after applying either *L. brevis* GQ423768 or *L. plantarum* GQ423760 topically for 21 days. *L. brevis*-treated wounds were found to start the healing process much faster than the *L. plantarum*-treated, and finally, on day 7, the wound healing percentage was smaller than in the former group. This finding was compatible with that of histology: although the reduction in the number of neutrophils was similar in both groups in relation to the control treatment, there was a sudden increase of neutrophils in the *L. plantarum* group on day 3. Regarding macrophages, a significant increase was clear from day 3, followed by a significant decrease on day 7 and thereafter in both probiotics-treated groups, as also occurred with the number of fibroblasts: a progressively significant increase was found in both groups from day 3 in relation to controls and only after day 14 was a reduction observed in relation to controls [30].

Incubation of HaCaT keratinocytes with the parabiotic *L. brevis* SGL 12 lysate resulted in a significant increase in cell proliferation compared to the control culture, the number of cells having increased 2.7-fold at 24 h post-treatment. However, when the parabiotic *L. brevis* SGL 12 lysate was applied on a scratched HaCaT keratinocyte monolayer culture, no cell migration was prominent, the findings suggesting that *L. brevis* SGL 12 acts by means of cell proliferation but not cell migration. Regarding pro-inflammatory chemokines release after keratinocyte exposure to the parabiotic *L. brevis* SGL 12, it was found that it reduces the secretion of RANTES 1.2 fold (*p* = 0.0252) and that of IL-8 1.5 fold (*p*  =  0.0358) in relation to controls. Finally, on a co-culture of *S. aureus*-infected keratinocytes with viable *L. brevis*—but not its lysate—over a 24 h period, it was found the *L. brevis* exerts a strong inhibition of *S. aureus* growth (1.8-log_10_ reduction, *p* < 0.0001) [38], suggesting again that its anti-microbial properties may have an indirect effect on wound healing.

### 2.5. Lacticaseibacillus casei (Formerly Lactobacillus casei) 

The probiotic bacteria *Lacticaseibacillus casei* 324 m was applied on a scratched monolayer culture of the immortalized human gingival epithelial cell line OBA-9, which was infected with *Porphyromonas gingivalis* W83. *Porphyromonas gingivalis* is a known pathogen that inhibits the release of CXCL8 by gingival epithelial cells and thus reduces their proliferation by means of releasing toxins or enzymes that are active against many substrates, including collagen, epithelial cells, and fibroblasts. *L. casei* 324 m was found to significantly accelerate the percentage of re-epithelialization and to control the harmful effects of *P. gingivalis* by increasing the concentration and up-regulation of CXCL8 expression in relation to controls. Again, although this anti-microbial effect may not have a direct role in wound healing, it seems to play a significant role in this process. Moreover, *L. casei* increases the number of receptors CXCR1 and CXCR2 on a gingival cell culture by up to 70%, but the inhibition of these receptors blocks the CXCL8-induced re-epithelialization [48].

The parabiotic lysate of *L. casei* SGL 15, applied on a scratched HaCaT keratinocyte monolayer culture, showed no migration process, while incubation resulted in a strong increase in proliferation in relation to controls at 24 h post-treatment. The number of keratinocytes increased 3.7-fold, suggesting that the parabiotic *L. casei*, similar to *L. brevis*, promotes epithelialization through cell proliferation and not migration. The response of HaCaT cells to the parabiotic *L. casei* lysates, in terms of the release of pro-inflammatory chemokines, was poor: *L. casei* SGL 15 was found to decrease the release of only IL-8 2.1-fold and had no other effect, such as that of MCP-1 RANTES, IP-10, and cytokines IL-1 alpha and IL-6 [38]. 

Finally, the *L. casei* Shirota was applied topically or given orally to rats for 3 or 7 days after induction of a traumatic/burn ulcer on the labial fornix incisive inferior (mandibular labial mucosa) to investigate the stimulation and proliferation of fibroblast cells and blood vessel formation during the healing process. Hematoxylin-eosin staining revealed a significantly increased number of fibroblasts and blood vessels on day 8 in relation to day 3, with the number being higher after topical treatment for 7 days than oral treatment, a finding supporting the option of topical use and long-term application of probiotic treatment [49].

### 2.6. Limosilactobacillus reuteri (Formerly Lactobacillus reuteri) 

The parabiotic lysate of *Limosilactobacillus reuteri* ATCC55730 applied on a scratched keratinocyte monolayer culture was found to significantly (4-fold) increase the number of migrated keratinocytes in relation to the control treatment. Similarly, incubation of keratinocytes with the parabiotic *L. reuteri* lysate resulted in significant proliferation (2-fold) of the number of cells in relation to controls, as well as to the parabiotic *L. rhamnosus* lysate. However, when Mitomycin-C, a known inhibitor of cell proliferation, was added to the scratched monolayer treated with *L. reuteri*, only 52% re-epithelialization occurred, in contrast to *L. rhamnosus* which achieved a 90% re-epithelialization, despite the presence of Mitomycin-C. This finding suggests that the dominant mechanism for *L. reuteri* is the stimulation of proliferation rather than migration [45].

*L. reuteri* DSM 17938 in a base of Eucerin ointment was applied topically daily for 15 consecutive days on full-thickness excisional wounds performed on the dorsum of rats. *L. reuteri*-treated rats exhibited a significant reduction in wound area diameter after day 5, findings also confirmed by histology: reduced inflammation, accelerated epithelialization, and collagen deposition in relation to control and placebo (Eucerin ointment only) treatment being more pronounced on days 10 and 15. Additionally, tissue myeloperoxidase activity was also found significantly decreased on days 5 and 10, suggesting that *L. reuteri* DSM 17938 exerts antioxidant properties, probably against the free radicals produced by neutrophils in the early phase of healing [50].

In a very different study in terms of study material, the healing properties of *L. reuteri* ATCC 11284 were first tested in a scratched culture of C57BL/6 mice gingiva mesenchymal stem cells (GMSCs). *L. reuteri*, in a dose of 50 μg/mL, was found to increase expression of stem cell markers (SOX2, OCT4, NANOG), as assessed by means of real-time RT-PCR; augment the alkaline phosphatase activity of GMSCs after osteogenic induction for 5 days; and enhance mineralization of GMSCs after osteogenic induction for 3 weeks (Alizarin red staining). Further, *L. reuteri* was found to promote the modulation of osteogenic differentiation (expression of transcription factors RUX2, OSX, and OCN) and the proliferation potential of GMSCs, while inhibiting the adipogenic differentiation of GMSCs. Looking for the underlined cellular mechanisms, researchers concluded that *L. reuteri* enhances the GMSCs scratch migration via the PI3K/AKT/β-catenin/TGF-β1/MMP-1 pathway. They then performed a full-thickness bilateral excisional wound on the mesial gingival of the maxillary first molar in mice, where they injected *L. reuteri* extracts or a placebo. The wound length was observed significantly reduced after 7 days of treatment, while, in parallel, matrix metalloproteinase (MMP)-1, known to facilitate the migration of fibroblasts and keratinocytes and degrade collagen types I and III to remodel the scar tissue, significantly increased [51].

In a human epithelial keratinocyte cell culture infected with 10^8^ cfu/mL *Staphylococcus aureus*—a dose that leads to 69.5% cell death within 24 h—the probiotic *L. reuteri* ATCC55730 was applied in order to investigate whether it can inhibit *Staphylococcus aureus* infection. *L. reuteri* was found to increase the cell survival rate (46.9% deaths) only if administered before or at the same time as *S. aureus* and only as live bacteria. Its protective effect was independent of inhibitory substances such as lactic acid but inhibited the adherence of *S. aureus* to keratinocytes by competitive exclusion, that is, by blocking the α5β1 integrin, which is also necessary for *S. aureus* to adhere to keratinocytes. In other words, *L. reuteri* should prevent skin infections when given as topical prophylactic treatment by means of inhibiting the adherence of *S. aureus* in its binding sites [44].

### 2.7. Limosilactobacillus fermentum (Formerly Lactobacillus fermentum) 

When the parabiotic *L. fermentum* lysate was added to a scratched keratinocyte monolayer, it was found to significantly reduce the re-epithelialization process by 40% in relation to controls [33]; the phenomenon is likely related to the reduction of keratinocyte viability induced by the parabiotic of *L. fermentum* [52].

In a similar experiment in a scratched human keratinocyte cell line culture (HaCaT cells), the addition of *L. fermentum* SGL 10 lysate did not stimulate monolayer re-epithelialization. On the contrary, a transwell migration assay at 7 days was shown to enhance the migration capacity of HaCaT cells, while incubation of keratinocytes with the parabiotic *L. fermentum* lysate resulted in a significant (2.6-fold) increase in proliferation, compared to the control culture at 24 h post-treatment. Keratinocytes incubated for 12 h with the heat-inactivated *S. aureus* or *S. pyogenes* pathogens and then for another 12 h with *L. fermentum* lysate exhibited a higher viability than monolayers treated with the pathogen alone. Finally, 24 h exposure of HaCaT cells culture to *L. fermentum* lysate was found to significantly down-regulate the levels of pro-inflammatory chemokines IL-6 1.6-fold, of RANTES 1.5-fold, of IP-10 1.4-fold, and of cytokine IL-8 2.7-fold, all of which are directly involved in the inflammatory process [38]. Once again, it should be underlined that the anti-microbial effects of the parabiotic *L. fermentum,* although perhaps not having a direct effect on healing, significantly and indirectly contribute to the wound healing process. 

### 2.8. Saccharomyces boulardi

Based on the knowledge that insulin-like growth factors (IGFs) and epidermal growth factor (EGF) found in breast milk are responsible for facilitating the growth of gut epithelial cells, maturation of the intestinal mucosal barrier, and decreased bacterial translocation in preterm infants, Fordjour et al. investigated the effect of *Saccharomyces boulardi* as an adjuvant to formula feeding in comparison to formula-only feeding in neonatal rats. Rats receiving the *S. boulardi*-enhanced feeding formula presented with moderately preserved bowel mucosa on day 4 compared to controls which exhibited severe inflammation associated with a significant increase in VEGF levels. Similar elevations in soluble VEGF receptor-1, IGF-I, and EGF were noted in all formula-feeding rats, while in those which received the *S. boulardi* supplementation, the values were more or less similar to that of maternally fed rats [53].

A preparation of postbiotic *S. boulardi* culture supernatant was used to investigate, both in vitro and in vivo, its role in regulating angiogenesis by means of VEGF-receptor (VEGFRs) activation, which thus modulates capillary vessel formation during inflammatory responses. First, postbiotic *S. boulardi* added (or control) to a human umbilical vein endothelial cell culture (HUVEC) treated with VEGF was found to significantly—in a dose-dependent manner—inhibit tubule formation, being a multi-step process involving cell adhesion, migration, differentiation, and growth. The effects of postbiotic *S. boulardi* on VEGF-induced angiogenesis were then tested in vivo in the ears of adult nude mice according to a previously developed angiogenesis model with the use of an adenovirus expressing VEGF-A. From day 7 until the end of postbiotic *S. boulardi* injection in the right ear on day 21, its inhibitory effect—reduced neo-angiogenesis—was clearly evident compared to the left ear; the results indicated that postbiotic *S. boulardi* inhibited VEGF-mediated angiogenesis. Finally, when given orally, *S. boulardi* was found to inhibit angiogenesis in a dextran sulphate sodium-induced colitis mouse model. The triple documented findings indicate that the probiotic yeast *S. boulardi* modulates angiogenesis and thus limits intestinal inflammation and promotes mucosal tissue repair by regulating VEGF-receptor signaling [54].

A human colonic subepithelial myofibroblast culture was stimulated by the probiotic yeast *S. boulardi* Unique-28 in order to investigate its effect on the expression of chemokines and wound-healing related factors, as well as their migratory rate. *S. boulardi* was found to down-regulate CCL5 mRNA in a dose of 10^2^ cfu/mL and to up-regulate it in a dose of 10^4^ cfu/mL. However, no other chemokine was found to be affected. As for protein chemokine production, although *S. boulardi* did not affect the expression of CXCL1 and CCL2, it slightly, but not statistically significantly, up-regulated the expression of CXCL8, while the CXCL10 remained undetectable. Collagen production was also dose-dependent: on a low dose, 10^2^ cfu/mL, the mRNAs of type ΙΙΙ were down-regulated, while on a high dose, 10^4^ cfu/mL, a statistically significant increase in its total protein production was prominent. Both doses led to the up-regulation of the Tissue Factor, but neither had any effect on the mRNA expression of collagen type I, fibronectin, and α-SMA. Finally, when *S. boulardi* was used at a dose of 10^4^ cfu/mL, it promoted the migration of myofibroblasts by 130.2% after 24 h and by 150.3% after 48 h [20].

### 2.9. Bifidobacterium longum

From a dermatology point of view, it has been demonstrated experimentally in an ex vivo human skin explant model and in a clinical study on volunteers that the administration of the parabiotic *B. longum* Reuter lysate improves sensitive skin. Ex vivo, a significant improvement in parameters relating to inflammation, such as a decrease in vasodilation, oedema, mast cell degranulation, and TNF-alpha release compared to placebo, was confirmed. Moreover, it significantly inhibited capsaicin-induced CGRP release by neurons. In the clinical application, skin sensitivity, assessed by the stinging test, and skin barrier recovery were found significantly decreased at the end of the treatment, which finally led to increased skin resistance against physical and chemical aggression, compared to the use of a control cream [55].

The re-epithelialization potential of the soluble fraction of the parabiotic *B. longum* BL-04 lysate was tested, among other Lactobacilli and Bifidobacteria genera, on a wounded (scratch test) human keratinocyte HaCaT cell-monolayer model. The results suggested that the parabiotic soluble fraction of *B. longum*—as well as those of *B. infantis* and *B. breve*—significantly inhibit the process, this being attributable to the significantly down-modulated nitric oxide synthase (NOS2) protein expression and activity and thus to a significant decrease in nitrite levels in culture medium, which also leads to decreased levels of IL-6 and IL-8, compared to an untreated culture [39]. 

The topical effect of *B. longum* UBBL-64 regarding the mRNA expression of pro-inflammatory, healing, and angiogenetic factors involved in the healing process in relation to L. plantarum UBLP-40 and *Lactobacillus rhamnosus* UBLR-58 was recently investigated on a standardized excisional wound in rats. It was found to significantly increase the mRNA expression of IL-1β only on days 2 and 15 and of IL-6 on days 2, 4, and 15, in relation to *L. plantarum*, which practically means *B. longum* appears to exert the smaller anti-inflammatory action in relation to *L. plantarum.* Regarding wound healing response, *B. longum* exerts a significantly increased mRNA expression of TGF-β and α-SMA but a less profound up-regulation of collagen type III mRNA expression compared to *L. plantarum.* However, it generally seems stronger in promoting the expression of angiogenic factors compared to *L. rhamnosus* since it significantly up-regulates EGF and VEGF mRNA expression; however, it also exhibits significantly higher values in VEGF mRNA expression in relation to *L. plantarum* up to day 4 and down-regulates thereafter [19].

### 2.10. Streptococcus thermophilus

The parabiotic soluble fraction from the lysate of *Streptococcus thermophilus*, applied on a scratched HaCaT keratinocyte monolayer culture, was found to significantly accelerate re-epithelialization in keratinocyte monolayers and exhibit a strong induction of keratinocyte migration and proliferation versus control treatment at both 20 h and 28 h from injury. The rapid repair of keratinocyte scratch assay after treatment with the parabiotic soluble fraction of *S. thermophilus* lysate was attributed to its ability to increase the nitrate levels in the culture medium in relation to the control since a marked and significant up-regulation of nitric oxide synthase 2 (NOS2) expression was demonstrated by immunoblotting data and by nitrite level assay [39].

Normal human dermal fibroblasts first exposed to TGF-β1 to establish an abnormal activation of myofibroblasts—resembling pathological scarring—were then treated with parabiotic lysate of *S. thermophilus* DSM 24731 strain to test whether it could antagonize the fibrogenic effects of TGF-β1. The parabiotic *Streptococcus thermophilus* treatment was found to significantly reduce the TGF-β1-induced cell proliferation, migration, and myo-differentiation in a time- and concentration-dependent manner. In addition, the treatment with parabiotic lysate reduced the α-SMA, fibronectin, and collagen-I expression levels and affected the collagen contraction ability of activated dermal fibroblasts. Moreover, parabiotic *S. thermophilus* targeted the TGF-β1 signaling, reducing Smad2/3 activation, TGF-β1 mRNA level, and β-catenin expression through the up-regulation of PPARγ [56]. 

### 2.11. Lacticaseibacillus paracasei (Formerly Lactobacillus paracasei) 

When applied to human foreskin fibroblasts (Hs68 cells), the heat-killed *L. paracasei* GMNL-653 parabiotic preparation was found to promote collagen synthesis and up-regulate the gene expression of serine palmitoyl-transferase small subunit A compared to control treatment. Furthermore, it exerts an inhibitory effect on the expression of αSMA, Smad2/3, and phosphorylated Smad2, as indexes of TGF-β induced myofibroblast differentiation, in a dose-dependent manner and has a positive effect—up-regulation of MMP1 synthesis. However, when heat-killed *L. paracasei* GMNL-653 parabiotic was applied to a mouse tail wound, it displayed better activity in skin wound repair, significantly accelerating wound closure by days 3 and 15 in relation to heat-killed cells of *L. plantarum* GMNL-6 parabiotic, although both formulas achieved wound healing by day 20 as opposed to control-treated mice, showing a 20% healing deficit. Additionally, *L. paracasei* parabiotic was found to increase MMP-1 expression in the early phase of healing (day 5) but gradually decrease in the next phase (day 9) in parallel with the presence of structured collagen fibers in the dermis of treated wounds as well as decreased expression of αSMA. All the above strongly indicate that the parabiotic of *L. paracasei* GMNL-653 has not only apparent wound healing ability but also prevents excessive skin fibrosis, all mediated by the functional ingredient of the probiotic cell wall, the lipoteichoic acid [40]. 

Incubation of HaCaT keratinocytes with the parabiotic of *L. paracasei* SGL 04 lysate resulted in a significant increase of proliferation relative to the control culture at 24 h post-treatment, and the number of cells increased 3.2-fold. When parabiotic *L. paracasei* was applied after a 12 h exposure of the keratinocyte culture to S. aureus or S. pyogenes, the keratinocytes shared a higher viability in relation to the non-treated controls. Pro-inflammatory chemokines released after keratinocytes exposure to parabiotic *L. paracasei* decreased the release of MCP-1 1.3-fold (*p* = 0.002), RANTES 1.8-fold (*p* = 0.017), and IL-8 2.3-fold (*p* = 0.003) in relation to controls. Interestingly, MCP-1 was significantly decreased 1.6-fold when keratinocytes, stimulated first with S. pyogenes, were then treated with parabiotic *L. paracasei*. Finally, in a co-culture of S. pyogenes-infected keratinocytes with viable 10^8^ cfu/mL *L. paracasei*—but not its lysate—over a 24 h period, the *L. paracasei* was found to exert a strong inhibitory effect on S. pyogenes growth (3-log^10^ reduction, *p* = 0.01) [38].

## 3. Discussion

Trauma of any kind (and etiology) is considered damage to tissue integrity. Most commonly, with the exception of blunt, internal organ trauma, it includes the skin epithelium, which is the host barrier to the external attack and is largely responsible for body homeostasis. Today, based on current knowledge, teleologically, the most destructive consequence of trauma is the alteration in the beta diversity of resident microbiota. If severe, this can lead to an increased probability of infection, local and systemic, and even worse, the prolongation of healing time. 

Wound healing consists of a complex sequence of cellular signaling (with neutrophils, macrophages, fibroblasts, and epithelial cells mainly involved) and molecular signaling (with cytokines, chemokines, their inhibitors and receptors) and growth factors that orchestrate and promote resurfacing, reconstitution, and restoration of the tensile strength of the wounded area in an overlapping consequence of three characteristic phases: hemostasis/inflammation, cell proliferation, and remodeling or scar formation. The presence or absence of these cellular and molecular elements, at the right time and to the right degree, wonderfully coordinates the wound healing process [28,45,46]. A deep understanding of the wound healing process as a whole and the critical roles of each different cell and molecule involved will allow the development of more sophisticated therapies for wound repair, as well as the prevention of excessive fibrosis and bad scar formation.

In this review, we present and discuss the wound-healing effects of several probiotics, postbiotics, and parabiotics, which have been highlighted in many in vitro and in vivo animal studies. These favorable effects have been reported to take place during all three stages of the wound healing process, i.e., hemostasis/inflammation, cell proliferation, and remodeling or scar formation, and can be seen summarized in Table 1.

Given that tissue damage hinders the natural activity of the local microbiome and causes imbalance even at the phylum level by means of a shortage of specific genera and species, treatment with probiotics, postbiotics, or parabiotics has been suggested as an interesting, promising, therapeutic alternative; thus, there is a trend today towards local probiotic therapy through the restoration of the affected microbiome [18,20]. Hundreds of publications of both experiments based on scratched cell cultures and excisional wound models, as well as many clinical studies (less sophisticated than experimental ones), have highlighted the beneficial effects of probiotics and their products. More precisely, the topical application of probiotics or their lysates/extracts (parabiotics and postbiotics, respectively) has been shown overall to promote healing through the inhibition of the growth of pathogenic bacteria, the regulation of local inflammatory response and by interacting with epidermis cells [45]. Despite all these, it is overall very difficult to assess the precise role of probiotics in the healing process, knowing well that each strain has a separate, distinctive, and even multi-factorial feature of action. Here, we try to recognize the most documented points in the whole “healing” mechanism, wherein each of the most frequently involved probiotic species and strains are positively implicated.

Innate inflammation is the primary defense event against invasion by potential pathogens, initiated by injury-induced molecular signals, leading rapidly to the first recruitment of neutrophils and many pro-inflammatory cytokines, which attract further neutrophils and then facilitate the entry of monocytes. It is well known that delay, regardless of the cause, in debridement and subsequent phagocytosis of neutrophils leads to a chronic wound, which means the faster the debridement, the faster the healing. Probiotics seem to work toward that end in two steps: *L. plantarum* spp., as the USM8613, MTCC 2621, SGL07, and UBLP-40 have been recognized in many studies as being that which has the strongest anti-inflammatory action, not only by means of significantly up-regulating the pro-inflammatory factors IL-1β, IL-4, IL-6, IFN-γ, β-defensin and the mRNA expression of CXCL10 and CXCL8, which are seriously involved in neutrophil and macrophage recruitment, but also by earlier down-regulating IL-8, IL-6, MCP-1, and RANTES and up-regulating the anti-inflammatory IL-10 and others. These actions clearly document their marked anti-inflammatory effect, which is possibly involved in the reduction of hypertrophic wound scar formation [19,20,33,38,41]. Furthermore, earlier experiments, having documented a faster wound closure during the inflammatory phase (days 1 to 4), clearly suggested the anti-inflammatory properties of probiotics [18,34,35]. In the same manner, other probiotic bacteria, such as *L. rhamnosus* UBLR-58, *L. acidophilus* LA-5, *L. fermentum* SGL10, *L. brevis* GQ4237768, *L. brevis* SGL 12, *L. paracasei* SGL 04, and *B. longum* UBBL-64 have been found to exert anti-inflammatory action through the same mechanisms, but to a significantly lesser degree, compared mainly with *L. plantarum* [19,20,38,46,47]. Additionally, other probiotics, *L. plantarum* being the best, were found to exert antioxidant properties.

Then, monocytes, upon reaching the site of injury, are differentiated into macrophages, which further promote inflammation by releasing inflammatory factors and reactive oxygen species [57]. Lombardi et al. [39] have reported a significant up-regulation of nitric oxide synthase 2 [NOS2], leading to an increase of nitrate levels in the wound site, by *L. Plantarum* Lp-115, *L. acidophilus* and *Streptococcus thermophilus* DSM 24731, all very significantly less (*p* < 0.0001) than *L. plantarum*, while *B. longum* down-modulated the NOS2 production. In parallel, *L. reuteri* DSM 17938 was found to exert antioxidant properties since tissue myeloperoxidase was significantly decreased after treatment [50]. However, NOS2 is also involved in the regulation of endothelial cell recruitment to the ischemic wound; thus, these probiotics are also implicated in this process.

During the later stages of the inflammatory phase, in situ switching of M1 macrophages to the anti-inflammatory M2 phenotype occurs, stimulated by efferocytosis or changes in cytokines [57,58,59,60,61]. These M2 macrophages now express anti-inflammatory cytokines and arginase, along with various growth factors, which promote angiogenesis and cellular proliferation, migration, and differentiation of keratinocytes, fibroblasts, and epithelial cells. Recently, a speeding up of the switching process has been recognized in *L. plantarum*-treated patients [35].

Phagocytosis of apoptotic neutrophils by macrophages is an important event in the release of soluble factors, including transforming growth factor-beta (TGF-β), which plays a major role in the regulation of the formation and remodeling of granulation tissue, thus entering the healing process in the proliferating phase [61]. Practically, this second phase of wound healing begins with extensive activation of keratinocytes, fibroblasts, macrophages, and endothelial cells to promote wound closure, collagen deposition, and angiogenesis [57]. Chemoattractants released by macrophages attract fibroblasts and activate them to proliferate and modulate the production of matrix metalloproteinases (MMPs) and their inhibitors. *L. plantarum* GMNL-6, *L. reuteri* DSM17938, and *L. paracasei* GMNL-653 increased MMP-1 expression in the early stage (day-5) but then gradually decreased it (up to day 9), earlier than in controls, as also occurred with the *L. plantarum* USM8613 [33,40,51]. It is also reported that *L. acidophilus* treatment results in stellate rather than oval-shaped scars in controls, with the difference being attributable to the increased wound contraction [36].

Mature fibroblasts, after being differentiated into myofibroblasts, migrate towards the granulation tissue and, driven by TGF-β, are ready to express α-smooth muscle actin (α-SMA), which contracts the wound. *L. plantarum* GMNL-6, *L. paracasei* GMNL-653, and *S. thermophilus* DSM 24731 were found to exert an earlier, dose-dependent inhibitory effect on α-SMA expression, in parallel with the presence of structured collagen fiber, probably contributing to the prevention of excessive skin fibrosis [40,56].

Additionally, myofibroblasts, stimulated by TGF-β, also initiate collagen synthesis and collagen deposition, the immature collagen III being replaced by collagen I, which has a higher tissue strength [58,62,63]. *L. plantarum* UBLP-40 was found to initiate a significant induction of collagen III mRNA expression from day 2 while reducing the TGF-β expression by day 8 [19,20], as also occurs with Streptococcus thermophilus DSM 24731 in a time- and concentration-dependent manner [39]. However, B. longum UBBL-64 exerts a less profound up-regulation of collagen III mRNA expression in relation to *L. plantarum UBLP-40* [19]. Satish et al. [13] reported a significant inhibition of collagen I mRNA expression and an increased production of type III, suggesting that *L. plantarum* ATCC10241 can modulate not only the quantity but also the type of collagen synthesized in response to injury and infection, thus alleviating excessive scaring.

Keratinocytes, with the aid of MMP-1 and MMP-9, migrate and proliferate through the wound bed to meet keratinocytes from the opposite edges to form a thin epithelial layer [57,60,64]. *L. plantarum* USM8613 accelerated keratinocyte migration towards the center of the wound and complete re-epithelization by day 12 in relation to controls which achieve complete re-epithelization only by day 16 [33], while the *L. plantarum* SGL07 in keratinocyte monolayers exhibits a strong induction of both migration and proliferation [38]. Similar double action is exerted by *Streptococcus thermophilus* DSM 24731 [39], as well as *L. rhamnosus* GG ATCC 53103 and *L. reuteri*. Migration is the main action of *L. rhamnosus* and the proliferation that of *L. reuteri* [45]. Finally, *L. casei* 324 m and SGL 15 and *L. brevis* SGL 12 stimulate only cell proliferation [30,38,48], while *Lactobacillus fermentum* SGL 10 and *Saccharomyces boulardi* Unique-28 do not stimulate monolayer re-epithelialization but strongly enhance migration [20,38]. Incidentally, *L. acidophilus* LA-5 does not express a significant migratory effect on colonic subepithelial myofibroblasts [20].

Regarding angiogenesis, the process is initiated earlier than the controls upon neutrophil reduction. This, on its own, means that probiotics such as the *L. plantarum* spp., which exert strong anti-inflammatory properties, reduce the recruited neutrophils in the wound earlier, and thus angiogenesis begins earlier. Besides the vascular endothelial growth factor (VEGF) regulated by tissue hypoxia and the other pro-angiogenic factors, M2 macrophages release CXC-chemokines and MMPs, which directly promote angiogenesis through the differentiation of endothelial cells. In this phase, angiogenesis is considered essential to facilitate cell migration, meet the metabolic needs of the proliferating wound cells, and improve the synthesis of extracellular matrix compounds [57,58,59,65]. It has also previously been mentioned that some *Lactobacilli* spps, such as *L. Plantarum* Lp-115 [39], have been reported to exert a significant up-regulation of NOS2, which facilitate endothelial cell recruitment to the ischemic wound. In a recent publication, Panagiotou et al. [19] reported that *L. plantarum* UBLP-40 presents a step-by-step up-regulation of VEGF mRNA up to day 8 when it exceeds the control value and then down-regulates. However, *L. rhamnosus* UBLR-58 is presented as more active, significantly up-regulating both EGF and VEGF mRNA expression from day 2, with no further fluctuation thereafter. In the same manner, Moreira et al. found an increased blood vessel density and increased VEGF levels in *Lactobacillus rhamnosus* CGMCC 1.3724 LPR-treated mice, practically confirmed by an increase in blood flow assessed by laser Doppler velocimetry [43]. At the clinical level, the topical application of *L. plantarum* ATCC 10241 in chronic diabetic foot ulcers demonstrated a significantly increased angiogenesis after 21 days [35], while mice treated with *Lactiplantibacillus plantarum MTCC 2621* revealed at histopathology an enhanced increase of angiogenesis on day 7 [41]. Speaking of extremes, *B. longum* UBBL-64 exerts the strongest expression of VEGF mRNA compared to both *L. rhamnosus* UBLR-58 and much more to *L. plantarum* UBLP-40 up to day 8 and then down-regulating, progressively [19]; on the other hand, *S. boulardi* was found, in a dose-dependent manner, to significantly inhibit VEGF-induced angiogenesis, both in cell culture and in vivo [54].

Remodeling brings the wound-healing process to an end. In this stage, lasting several weeks after injury, the granulating tissue matures and becomes a scar of high tensile strength, in parallel exerting higher resistance and flexibility. Matrix metalloproteases break down collagen III and replace it with collagen I, the primary collagen subtype in scar tissue, which further re-organizes into parallel fibrils, forming a low cellularity scar [58,62,64,66]. Since our data on probiotics is mainly based on cell cultures and excisional wound models in experimental animals, there is no published information on the long-term effects of probiotics in the healing process. Nevertheless, what remains extremely useful so far is that probiotics, each in its own way and to a different degree, contribute to faster healing; the speeding-up process teleologically working towards better maintaining the homeostasis of the injured organism and thus preventing the entry of bacteria into the open wound.

Finally, many studies analyzed in this review investigated the effect of postbiotics, meaning the products resulting from non-viable probiotics, on wound healing. One possible explanation for choosing to treat wounds with postbiotics is the concern for possible infections due to the presence of live probiotics. Although this has been considered a possible risk for immunosuppressive patients [67], probiotic administration seems to no longer raise any significant safety concerns, as many studies report none or very few adverse effects from probiotic treatment [68].

## 4. Review Criteria

We searched for original articles exclusively, using the keywords “*probiotics*” AND “*trauma* OR *wound healing*” in PubMed and MedLine, with no restriction on publication date. The search terms were classified per probiotic genera, the only limiting criteria being at least two references for each probiotic included. 

## 5. Conclusions

Our assessment of the most studied probiotics and their downstream by-products led us to conclude that all species, in different ways, speed up the healing process, as follows:Almost all lactobacilli, but most significantly *L. plantarum*, exert a high pro-inflammatory action at first, resulting in the rapid debridement of the wounded area by neutrophils followed by monocytes/macrophages;*L. plantarum* and others (*L. acidophilus*, *Streptococcus thermophilus*) also exert antioxidant properties;*L. plantarum* then switches the earlier macrophages phenotype from M1, pro-inflammatory, to M2, anti-inflammatory, promoting angiogenesis, migration, and proliferation of keratinocytes and fibroblasts;*L. plantarum*, *L. reuteri*, *L. acidophilus*, and *L. paracasei* increase MMP-1 expression in the early stage and then reduce it to achieve better wound scarring;*L. plantarum*, *L. paracasei*, and *Streptococcus thermophilus* exert an inhibitory effect on α-SMA production, thus preventing excessive fibrosis;*L. plantarum*, *Streptococcus thermophilus*, and, to a lesser extent, *B. longum*, through TGF-β, initiate earlier collagen III synthesis and deposition, then replace the immature collagen III with the type I. *L. plantarum* controls both the quality and quantity of collagen deposition to alleviate excessive scarring;Almost all *lactobacilli* accelerate keratinocyte migration and proliferation to a different degree, and some support only proliferation or migration;*B. longum* and, to a lesser degree, *L. rhamnosus* and *L. plantarum* significantly increase blood vessel density by up-regulating VEGF and/or EGF expression. However, *Saccharomyces boulardi* inhibits VEGF-induced angiogenesis.

## Figures and Tables

**Figure 1 nutrients-15-03055-f001:**
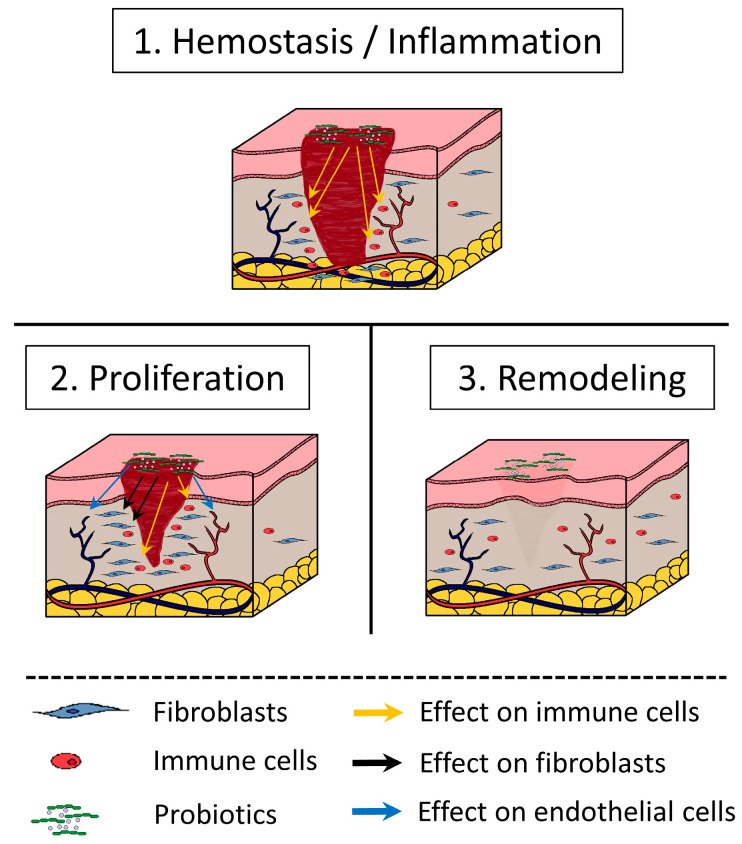
The effect of probiotics on wound healing stages.

**Table 1 nutrients-15-03055-t001:** The effects of probiotics, postbiotics, and parabiotics on the wound-healing process. ↑ means increase or speed-up, ↓ means decrease.

Strain	Type	In Vitro/In Vivo/Clinical Studies	Effect	Study
*L. plantarum*	Probiotic	gastric ulcers (rats)	↑ healing ↓ inflammation	[29]
*L. plantarum*	Probiotic	excisional skin wounds (rats)	↑ healing ↓ inflammation	[30]
*L. plantarum*	Probiotic	excisional skin wounds (rats)	↑ healing ↓ inflammation	[31]
*L. plantarum 299v*	Probiotic and postbiotic	skin burns (rats)	↑ healing ↓ infection	[32]
*L. plantarum*	Probiotic	skin burns (rabbits)	↓ scarring ↓ infection	[13]
*L. plantarum*	Postbiotic	porcine skin model	↑ healing ↓ infection	[33]
*L. plantarum*	Probiotic	diabetic wounds (rats)	↑ healing Regulation of inflammation	[34]
*L. plantarum*	Probiotic	diabetic foot (human)	↑ healing ↓ infection	[35]
*L. plantarum*	Probiotic	excisional skin wounds (rats)	No significant effect	[36]
*L. plantarum*	Probiotic	Caco-2 cell culture and 5-FU mucocitis (mice)	↓ inflammation ↓ infection	[37]
*L. plantarum* UBLP-40	Probiotic	excisional skin wounds (rats)	↓ inflammation	[19]
*L. plantarum* UBLP-40	Probiotic	excisional skin wounds (rats)	↑ healing ↓ inflammation	[18]
*L. plantarum* SGL07	Parabiotic	keratinocytes culture	↑ proliferation ↑ migration ↓ inflammation ↓ infection	[38]
*L. plantarum * Lp-115	Parabiotic	HaCaT monolayer culture	↑re-epithelization	[39]
*L. plantarum* GMNL-6	Parabiotic	Hs68 fibroblast cell culture and excisional skin wounds (rats)	↑ healing ↓ scarring	[40]
*L. plantarum MTCC 2621*	Postbiotic	A549 cell culture and excisional skin wounds (mice)	↑ healing ↓ infection	[41]
*L. plantarum* UBLP 40	Probiotic	intestinal myofibroblast culture	Regulation of inflammation ↑ migration	[20]
*L. rhamnosus* GG ATC 53103	Postbiotic	intestinal epithelial cell culture	Protection from oxidant stress	[42]
*L. rhamnosus CGMCC1.3724 LPR*	Probiotic	excisional skin wounds (mice)	↑ healing ↑ angiogenesis ↓ scarring ↓inflammation	[43]
*L. rhamnosus* GGATCC53103	Probiotic	keratinocytes culture	↓ infection	[44]
*L. rhamnosus* GG	Probiotic & Parabiotic	keratinocytes culture	↓ infection	[52]
*L. rhamnosus* UBLR-58	Probiotic	excisional skin wounds (rats)	↑ healing ↑ angiogenesis	[19]
*L. rhamnosus* GGATCC53103	Parabiotic	keratinocytes culture	↑ proliferation ↑ migration	[45]
*L. rhamnosus* LR-32	Probiotic	CD14 + monocytes culture	Possible antibacterial effect	[46]
*L. reuteri* ATCC55730	Parabiotic	keratinocytes culture	↑ proliferation	[45]
*L. reuteri * DSM 17938	Probiotic	keratinocytes culture	↑ healing	[50]
*L. reuteri * ATCC 11284	Parabiotic	mesenchymal stem cells culture	↑ proliferation ↑ migration	[51]
*L. reuteri* ATCC55730	Probiotic	keratinocytes culture	↓ infection	[44]
*L. acidophilus*	Probiotic	excisional skin wounds (rats)	↑ healing	[36]
*L. acidophilus*	Parabiotic	HaCaT monolayer culture	↑re- epithelization	[39]
*L. acidophilus * LA-5	Probiotic	CD14 + monocytes culture	Possible antibacterial effect	[46]
*L. acidophilus * LA-5	Probiotic	intestinal myofibroblast cell culture	Regulation of inflammation	[20]
*L. casei* 324 m	Probiotic	gingival epithelial cells culture	↑ proliferation ↓ infection	[48]
*L. casei* SGL 15	Parabiotic	keratinocytes culture	↑ proliferation ↑ migration ↓ inflammation ↓ infection	[38]
*L. casei shirota*	Probiotic	excisional skin wounds (mice)	↑ healing ↑ angiogenesis	[49]
*L. brevis* GQ423768	Probiotic	excisional skin wounds (rats)	↑ healing ↓ inflammation	[47]
*L. brevis * SGL 12	Parabiotic	keratinocytes culture	↑ proliferation ↑ migration ↓inflammation ↓ infection	[38]
*L. fermentum*	Parabiotic	keratinocytes culture	↓ re- epithelization	[45]
*L. fermentum * SGL 10	Parabiotic	keratinocytes culture	↑ proliferation ↑ migration ↓inflammation ↓ infection	[38]
*B. longum reuter*	Parabiotic	dermal skin explant (human) nerve cell culture	↓ skin sensitivity	[55]
*B. longum* BL-04	Parabiotic	HaCaT monolayer culture	↓re-epithelization	[39]
*B. longum UBBL-64*	Probiotic	excisional skin wounds (rats)	↑ healing ↑ angiogenesis	[19]
*S. boulardi*	Probiotic	gut ulcer model (rats)	↓inflammation	[53]
*S. boulardi*	Probiotic	human umbilical vein endothelial cells culture	↑ healing ↑ angiogenesis ↓inflammation	[54]
*S. boulardi*	Probiotic	intestinal myofibroblast cell culture	Regulation of inflammation ↑ migration	[20]
* L. paracasei * GMNL-653	Parabiotic	Hs68 fibroblast cell culture and excisional skin wounds (mice)	↑ healing ↓ scarring	[40]
*L. paracasei * SGL 04	Parabiotic	keratinocytes culture	↑ proliferation ↑ migration ↓inflammation ↓ infection	[38]
*S. thermophilus*	Parabiotic	HaCaT monolayer culture	↑re- epithelization	[39]
*S. thermophilus* DSM 24731	Parabiotic	dermal fibroblast cell culture	↓ fibrosis	[56]

## Data Availability

This study did not report any data.

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
