# Peer review of "Looking for the Ideal Probiotic Healing Regime"

_nutrients, 2023, doi:10.3390/nu15133055_

Round 1

Reviewer 1 Report

General comments

In this manuscript, the authors reviewed the role of different probiotic species on wound healing.

The work provides potentially interesting and valuable information on an area of clinical importance, even though it should be emphasized that studies have so far been performed only in vitro or in animal models.

However, some parts of the manuscript need modification for better clarity.

Specifically, in my view the different phases of the wound healing process, which is the focus of the paper, should be moved from the Discussion to the Introduction. Also, any overlapping and redundancy between the Introduction and Discussion sections should be avoided.

When describing the role of probiotics on wound healing, the authors often mention their preventive effects towards wound infections: however, prevention of wound infection does not seem to have an active role in the wound healing process, but it is rather a distinct effect, and this should be clarified in the text.

In addition, in the title and in the text the authors generally refer to “probiotics”: however, most of the reported studies make use of “probiotic lysates” or “heat-killed” probiotics or “cell-free supernatants”, all of which fall in the ”postbiotics” definition (see for reference Vinderola G, Sanders ME, Salminen S. The concept of postbiotics. Foods. 2022 Apr 8;11(8):1077. doi: 10.3390/foods11081077). The reason of using postbiotics rather than probiotics in those studies could be discussed by the authors: for instance, it would be interesting to know whether there might be risk of sepsis related to the direct use of live  microorganisms (the probiotics) on a wound. Has this possibility ever been investigated in animal models?

Finally, since there is a large heterogeneity among the many studies cited by the authors, it would be important to detail for each study: the probiotic strain and the dose used (even for postbiotics), the target cell line, or animal model and the administration route, and the assessment of wound healing.  

Specific comments

Page 1, lines 29-34. The use of the “microbiota” and “microbiome” terms should be consistent.

Page 1, lines 39-42. Biofilm formation, growth of pathogenic microorganisms, bacterial colonization, clinical sepsis are very similar events and should not be referred to “most simple” and “most complex”.

Page 2, lines 60-61. To my knowledge, whether multi-strain probiotic formula are more effective than single-strain probiotics remains to be clarified (see for reference: McFarland LV. Efficacy of Single-Strain Probiotics Versus Multi-Strain Mixtures: Systematic Review of Strain and Disease Specificity. Dig Dis Sci. 2021 Mar;66(3):694-704. doi: 10.1007/s10620-020-06244-z.)

Page 2, lines 62-63. Please, provide reference(s) on probiotics inhibiting one another.

Page 2, line 71. Please, provide reference(s) for the new taxonomic nomenclatures substituting the former ones for the various probiotic species mentioned here and in the other subtitles later in the text.

Page 2, lines 81-82. Specify whether it was oral administration of L. plantarum.

Page 2, lines 92-93. Specify whether it was topical administration of L. plantarum.

Page 4, lines 164-166. The authors state that the dose of 106 CFU/g orally administered to rats is low: could they indicate what are the mean effective doses in similar or other studies?

Page 4, line 172. Please, explain the meaning of 5-FU.

Page 5, line 223. Specify that it was heat-killed L. plantarum GMNL-6.

Page 8, lines 369-370. Please, specify which animal model was used.

Page 9, lines 456-458. Please, specify which animal model was used.

Page 10, lines 478-479. Please, specify whether it was viable L. fermentum SGL10 or its lysate.

Page 12, line 593. Please, specify whether it was viable L. plantarum GMNL-653 or the heat-killed cells.

Page 13, lines 639-641. From this review, it is clear that no human clinical studies have been performed in which probiotics are directly used on wounds. Even if minimal, there might be a potential risk for sepsis due to probiotic bacteria, as they are by definition live microorganisms. Obviously, using live microorganisms is not like using antibiotics, as different risks are implied. Based on these considerations, I would suggest to remove this sentence.

Author Response

REVIEWER 1

In this manuscript, the authors reviewed the role of different probiotic species on wound healing.The work provides potentially interesting and valuable information on an area of clinical importance, even though it should be emphasized that studies have so far been performed only in vitro or in animal models.

Authors’ response:

We would like to thank the reviewer for taking the time to review our manuscript. Indeed most of the studies have been done only in vitro or in vivo and this has been now highlighted at the end of the Introduction.

However, some parts of the manuscript need modification for better clarity. Specifically, in my view the different phases of the wound healing process, which is the focus of the paper, should be moved from the Discussion to the Introduction. Also, any overlapping and redundancy between the Introduction and Discussion sections should be avoided.

Authors’ response:

We change the Introduction section by adding, on the 2nd paragraph, line 40, a short description of wound healing process. “… in an overlapping consequence of three phases: Initially, the blood clot plugged on the wound bottom acts as a temporary barrier to external invasion while, in parallel coagulation factors indirectly involved in neutrophils and macrophages migration to initiate the inflammatory phase – practically to clear the field. Upon this accomplishment, macrophages turned into M2 anti-inflammatory ones, to guide tissue regeneration, in parallel with fibroblasts migration. Consequently, a granulation tissue consisting of macrophages, fibroblasts, connective tissue and new blood vessels is developed to replace the initial blood clot and, then, fibroblasts differentiate into α-SMA-expressing myofibroblasts to increase the wound stiffness, terminating thus tissue remodeling. Finally, wound cellularity decreases and collagen type III in the extracellular matrix is replaced by collagen type I to form the scar.” In addition, we also changed the beginning of the Discussion in order to avoid overlapping and redundancy with the Introduction.

When describing the role of probiotics on wound healing, the authors often mention their preventive effects towards wound infections: however, prevention of wound infection does not seem to have an active role in the wound healing process, but it is rather a distinct effect, and this should be clarified in the text.

Authors’ response:

Although prevention of wound infection may not have a direct effect on wound healing, we believe that it plays a significant indirect role, as ultimately, an infected wound would require more time to heal than a non-infected one. Nevertheless, we have clarified this in the text.

In addition, in the title, and in the text the authors generally refer to “probiotics”: however, most of the reported studies make use of “probiotic lysates” or “heat-killed” probiotics or “cell-free supernatants”, all of which fall in the ”postbiotics” definition (see for reference Vinderola G, Sanders ME, Salminen S. The concept of postbiotics. Foods. 2022 Apr 8;11(8):1077. doi: 10.3390/foods11081077).

Authors’ response:

As the reviewer requested, we have now mentioned the term postbiotics in the introduction.

The reason of using postbiotics rather than probiotics in those studies could be discussed by the authors: for instance, it would be interesting to know whether there might be risk of sepsis related to the direct use of live microorganisms (the probiotics) on a wound. Has this possibility ever been investigated in animal models? ).

Authors’ response:

Indeed, this possibility has been taken into consideration by several research groups, and, although live probiotic administration was first considered as a possible risk for immunosuppressive patients this has changed, as many studies report none to very few adverse effects from probiotics treatment. This comment has been now included in the Discussion.

Finally, since there is a large heterogeneity among the many studies cited by the authors, it would be important to detail for each study: the probiotic strain and the dose used (even for postbiotics), the target cell line, or animal model and the administration route, and the assessment of wound healing.  

Authors’ response:

As the reviewer suggested, we made a table describing the details of each study and we implemented it at the Discussion.

Specific comments:

Page 1, lines 29-34. The use of the “microbiota” and “microbiome” terms should be consistent.

Authors’ response:

We change it accordingly, Introduction, paragraph 1 “ Trauma, of whatever cause, either a clean surgical wound or a severe traumatic injury, leads to an alteration in the beta-diversity of resident microbiota due to the oxidative stress caused by both tissue injury and anesthesia. This alteration involves both the intestinal microbiota, being the most studied ― and definitely considered as a generalized reaction ― and the local microbiota, in this case, that of the skin [1-4].”

Page 1, lines 39-42. Biofilm formation, growth of pathogenic microorganisms, bacterial colonization, clinical sepsis are very similar events and should not be referred to “most simple” and “most complex”.

Authors’ response:

We shorten this sentence, in  the paragraph 2 “Although it is largely a "standard" process, in recent years there has been increasing certainty that the disruption of the local microbiome, as a consequence of trauma, exerts a positive involvement in the prolongation of the healing time; from the most simple, the biofilm formation and the growth of pathogenic microorganisms, to the most complex, the prolongation of the inflammatory phase and the transition from acute to delayed healing; and, even in clinical cases generally, to bacterial colonization, and possibly clinical sepsis [5,6]. “

Page 2, lines 60-61. To my knowledge, whether multi-strain probiotic formula are more effective than single-strain probiotics remains to be clarified (see for reference: McFarland LV. Efficacy of Single-Strain Probiotics Versus Multi-Strain Mixtures: Systematic Review of Strain and Disease Specificity. Dig Dis Sci. 2021 Mar;66(3):694-704. doi: 10.1007/s10620-020-06244-z.)

Authors’ response:

We change the phrase as follows: “With this knowledge, and given that a multi-probiotic formula seem exerting better results than a single strain [13-16] – although there are a few references supporting the opposite [ McFarland] − the pharmaceutical industry has embarked on a race…“

Page 2, lines 62-63. Please, provide reference(s) on probiotics inhibiting one another.

Authors’ response:

As the reviewer requested, we have now added the appropriate references.

Page 2, line 71. Please, provide reference(s) for the new taxonomic nomenclatures substituting the former ones for the various probiotic species mentioned here and in the other subtitles later in the text.

Authors’ response:

As the reviewer requested, we have added the appropriate reference.

Page 2, lines 81-82. Specify whether it was oral administration of L. plantarum.

Authors’ response:

The L. plantarum was given intragastrically and we have now mentioned it in the text.

Page 2, lines 92-93. Specify whether it was topical administration of L. plantarum.

Authors’ response:

In this reference, L. plantarum is topically administrated, as already mentioned in the text.

Page 4, lines 164-166. The authors state that the dose of 106 CFU/g orally administered to rats is low: could they indicate what are the mean effective doses in similar or other studies?

Authors’ response:

Similar studies discussed in the present review report doses of 1010 to 1011 [1,2]

Page 4, line 172. Please, explain the meaning of 5-FU.

Authors’ response:

5-fluorouracil – has been added to the text.

Page 5, line 223. Specify that it was heat-killed L. plantarum GMNL-6.

Authors’ response:

As the reviewer requested, we have specified it.

Page 8, lines 369-370. Please, specify which animal model was used.

Authors’ response:

It refers to a rat-model. We have now mentioned it in the text.

Page 9, lines 456-458. Please, specify which animal model was used.

Authors’ response:

It refers to mice model.

Page 10, lines 478-479. Please, specify whether it was viable L. fermentum SGL10 or its lysate.

Authors’ response:

It is referred to L. fermentum lysate.

Page 12, line 593. Please, specify whether it was viable L. plantarum GMNL-653 or the heat-killed cells.

Authors’ response:

It is referred to heat-killed cells.

Page 13, lines 639-641. From this review, it is clear that no human clinical studies have been performed in which probiotics are directly used on wounds. Even if minimal, there might be a potential risk for sepsis due to probiotic bacteria, as they are by definition live microorganisms. Obviously, using live microorganisms is not like using antibiotics, as different risks are implied. Based on these considerations, I would suggest to remove this sentence.

Authors’ response:

As the reviewer requested, we have removed this sentence [Discussion, Paragraph 4]

Reviewer 2 Report

Major revision is needed. Please see the suggestions below. 

General comments-

The authors should explain at the beginning what the stages of wound healing are (although this explanation is included in the discussion, it should still be mentioned at the beginning, for easy follow-up), and only then explain how probiotics fit into the topic of wound healing. Is it all about reducing the inflammatory process, or is there something else? It is also necessary to show what is the goal when probiotics are given orally, and what is the goal when they are applied to the skin. Separate manuscript that includes: in vivo, on cells cultures-in vitro assays and clinical studies. Present that in tables. In general, authors should provide some overview tables and diagrams to make the data easier to follow. What was the methodology for selecting the manuscripts, from which years were the papers selected, from which database? Is there a history of using natural preparations based on probiotics for wound healing, this should be interesting to mention. Which drugs are most often combined with probiotics that are used for wound healing? Next, the authors should consider to present what beneficial probiotics are found on human skin. Aren't there already commercially available creams, serums for regeneration, for faster healing of wounds, for sunburns... deodorants that contain probiotics? The only thing that can be seen that the authors presented is the Eucerin product with L. reuteri. More available products need to be listed. From the title it is expected that the work will be focused more on the skin regeneration, but this paper mainly deals with probiotics that suppress biofilms of pathogenic microbes-i.e., antibiofilm effect and also inflammation.

Specific suggestions-

Key words Lactobacillus rhamnosus, L. reuteri, L. casei

It is necessary to change the names according to the newer nomenclature

L72 L. plantarum is the most studied probiotic

Provide a reference

L169 isolated from Brazil

Are these human-derived isolates or is it isolated from Brazilian food?

L214 Tasai et al.,

It is necessary to write the year of publication

L214 heat killed...

Please refer to probiotic, postbiotic, parabiotic... The authors should pay attention on difference between viable probiotic bacteria, inactivated ones, and their metabolites, and call them by their proper names (probiotic, postbiotic, parabiotic).

L285 L. Reuteri

reuteri should be in lower case

L344 Vale et al

Transfer 34 reference after name

L356 Tarapatzi et al

Reference number is missing

L604 It is necessary to write S. aureus in italics, generally pay attention to italics in microbe names

Author Response

REVIEWER 2

The authors should explain at the beginning what the stages of wound healing are (although this explanation is included in the discussion, it should still be mentioned at the beginning, for easy follow-up), and only then explain how probiotics fit into the topic of wound healing. Is it all about reducing the inflammatory process, or is there something else?

Authors’ response:

We have added information of wound healing at the Introduction, as requested by the reviewers.

It is also necessary to show what is the goal when probiotics are given orally, and what is the goal when they are applied to the skin. Separate manuscript that includes: in vivo, on cells cultures-in vitro assays and clinical studies. Present that in tables

Authors’ response:

We would like to thank the reviewer for their interesting suggestion. Nonetheless, we prefer to present the current literature based on the probiotic strains. We have included a Table at the beginning of the Discussion showing which type of probiotic/postbiotic/parabiotic and what experimental model were used and what was the effect.

In general, authors should provide some overview tables and diagrams to make the data easier to follow.

Authors’ response:

As reviewers suggested, we have presented these details in Table 1.

What was the methodology for selecting the manuscripts, from which years were the papers selected, from which database? Is there a history of using natural preparations based on probiotics for wound healing, this should be interesting to mention.

Authors’ response:

We added Review Criteria at the end of the text

Which drugs are most often combined with probiotics that are used for wound healing?.

Authors’ response:

We would like to thank the reviewers for their suggestion. Nonetheless, the scope of this review was to investigate the possible favorable effect of probiotics alone on wound healing. The effect of the co-administration of wound-healing-promoting drugs and probiotics would be an interesting topic for a future review that could further analyze the cellular and molecular mechanisms involved.

Aren't there already commercially available creams, serums for regeneration, for faster healing of wounds, for sunburns... deodorants that contain probiotics? The only thing that can be seen that the authors presented is the Eucerin product with L. reuteri. More available products need to be listed. 

Authors’ response:

We would like to thank the reviewer for their suggestion. Our review was mainly focused on discussing and analyzing the direct and indirect effects of pure probiotics or postbiotics on wound healing, in an effort to elucidate their possible healing mechanisms. The investigation of the role and effect of commercially available creams, serums, and deodorants supplemented with probiotics would be indeed an interesting idea for a future review, as all of these products, apart from the probiotics, usually include additional healing-promoting factors, such as vitamins, which also need to be analyzed and discussed.

From the title it is expected that the work will be focused more on the skin regeneration, but this paper mainly deals with probiotics that suppress biofilms of pathogenic microbes-i.e., antibiofilm effect and also inflammation.

Authors’ response:

As the healing process involves inflammation, proliferation and remodeling, this review was focused on the effect of probiotics on all of these stages. In addition, prevention of infections majorly affects the time of healing process and this is why we also discussed this type of studies.

Specific suggestions-

Key words Lactobacillus rhamnosus, L. reuteri, L. casei. It is necessary to change the names according to the newer nomenclature

Authors’ response:

It is revised as suggested by the reviewers.

L72 L. plantarum is the most studied probiotic. Provide a reference

Authors’ response:

As the reviewer requested, we added the appropriate reference.

L169 isolated from Brazil. Are these human-derived isolates or is it isolated from Brazilian food?

Authors’ response:

Coelho-Rocha et al isolated several different L. plantarum strains from Artisanal butter,  Manioc, Sugarcane juice, Cow’s milk, Beetroot, Pineaple, 3 days newborn faeces and Foal faeces. It is mentioned in their paper [1].

L214 Tasai et al. It is necessary to write the year of publication;

Authors’ response:

Ιt is mentioned on the reference list (2021).

L214 heat killed...

Authors’ response:

It is now changed to “parabiotic heat-killed L. plantarum GMNL-6”.

Please refer to probiotic, postbiotic, parabiotic... The authors should pay attention on difference between viable probiotic bacteria, inactivated ones, and their metabolites, and call them by their proper names (probiotic, postbiotic, parabiotic).

Authors’ response:

We would like to apologize. We have now included the terms postbiotics and parabiotics in the introduction and we have corrected the terms in the text.

L285 L. Reuteri; reuteri should be in lower case;

Authors’ response:

It is revised as suggested.

L344 Vale et al. Transfer 34 reference after name

Authors’ response:

It is revised as suggested.

L356 Tarapatzi et al. Reference number is missing

Authors’ response:

It is revised as suggested.

L604 It is necessary to write S. aureus in italics, generally pay attention to italics in microbe names;

Authors’ response:

It is revised as suggested.

Reviewer 3 Report

- In keywords, the names of microorganisms are not in italics.

- Isert a figure or a table to enrich the review.

- It is important that a review has at least 100 references. The manuscript has 56. I suggest increasing the number of references.

Author Response

REVIEWER 3

In keywords, the names of microorganisms are not in italics.

Authors’ response:

It is revised as suggested.

Insert a figure or a table to enrich the review.

We would like to thank the reviewer for their suggestion. We inserted a Figure and a Table .

It is important that a review has at least 100 references. The manuscript has 56. I suggest increasing the number of references.

Authors’ response:

It is a narrated review, exclusively based on the healing properties of some probiotics, postbiotics and parabiotics. We have stated in the aim of the study that ‘we included those only probiotics having at least 2 references on wound healing properties’, in order to provide more validated information. Thus, the references are 17 for L. plantarum, 8 for L. rhamnosus, 4 for L. Reuteri, 4 for L. acidophilus, 3 for Bifidobacterium longum, 3, for L. brevis, 3 for L. casei, 3 for S. boulardi, 2 for L. paracasei, and 2 for S. thermophilus.

Round 2

Reviewer 1 Report

The manuscript is now acceptable for publication.

Reviewer 2 Report

The manuscript has been significantly improved. Authors should pay attention to number of keywords. Maximum 10 keywords is allowed according to the journal's propositions.

L282 It is not common to refer to references in subheadings. L. rhamnosus appears for the first time in L218, you can introduce there that Lacticaseibacillus rhamnosus is formerly Lactobacillus rhamnosus. The same comment applies to other strains- insert references in the text.